# Rapid and Sensitive Detection of Toxigenic *Fusarium asiaticum* Integrating Recombinase Polymerase Amplification, CRISPR/Cas12a, and Lateral Flow Techniques

**DOI:** 10.3390/ijms241814134

**Published:** 2023-09-15

**Authors:** Jun Zhang, Xiaoyan Liang, Hao Zhang, Shumila Ishfaq, Kaifei Xi, Xueping Zhou, Xiuling Yang, Wei Guo

**Affiliations:** 1State Key Laboratory for Biology of Plant Diseases and Insect Pests, Institute of Plant Protection, Chinese Academy of Agricultural Sciences, Beijing 100193, China; cgjl_2016@126.com (J.Z.); zzhou@zju.edu.cn (X.Z.); 2Institute of Food Science and Technology, Chinese Academy of Agricultural Sciences, Key Laboratory of Agro-Products Quality and Safety Control in Storage and Transport Process, Ministry of Agriculture and Rural Affairs, Beijing 100193, China; liangxy_97@163.com (X.L.); 2020y90100028@caas.cn (S.I.);

**Keywords:** *Fusarium asiaticum*, recombinase polymerase amplification (RPA), Cas12a, lateral flow detection, maize and wheat

## Abstract

*Fusarium* head blight (FHB) is a global cereal disease caused by a complex of *Fusarium* species. Both *Fusarium graminearum* and *F. asiaticum* are the causal agents of FHB in China. *F. asiaticum* is the predominant species in the Middle–Lower Reaches of the Yangtze River (MLRYR) and southwest China. Therefore, detecting *F. asiaticum* in a timely manner is crucial for controlling the disease and preventing mycotoxins from entering the food chain. Here, we combined rapid genomic DNA extraction, recombinase polymerase amplification, Cas12a cleavage, and lateral flow detection techniques to develop a method for the rapid detection of *F. asiaticum*. The reaction conditions were optimized to provide a rapid, sensitive, and cost-effective method for *F. asiaticum* detection. The optimized method demonstrated exceptional specificity in detecting *F. asiaticum* while not detecting any of the 14 other *Fusarium* strains and 3 non-*Fusarium* species. Additionally, it could detect *F. asiaticum* DNA at concentrations as low as 20 ag/μL, allowing for the diagnosis of *F. asiaticum* infection in maize and wheat kernels even after 3 days of inoculation. The developed assay will provide an efficient and robust detection platform to accelerate plant pathogen detection.

## 1. Introduction

*Fusarium* head blight (FHB) is a crucial disease affecting wheat, barley, rice, maize, and other small grains worldwide, leading to significant economic losses [1]. Although a number of *Fusarium* species can cause FHB, the primary etiological agents of this disease belong to the *Fusarium graminearum* species complex (FGSC) of B trichothecene toxin producers. FGSC comprises at least 16 distinct species [2]. The most important member is the globally occurring species *F. graminearum* [3]. *F. asiaticum*, the primary pathogen of FHB in Asia, including China, Korea, Nepal, and Japan, is the second-most important species [4]. The Middle–Lower Reaches of the Yangtze River (MLRYR) and southwest China, where FHB is the most prevalent, are notably dominated by *F. asiaticum* [5,6,7,8]. *F. asiaticum* can produce type B trichothecenes such as NIV, 3-ADON, and 15-ADON that pose a significant threat to consumer and livestock health. The consumption of contaminated grains could induce immunotoxicity and cytotoxicity, which has become a potential threat to animals and humans [9]. Distinguishing the symptoms of *F. asiaticum* from other *Fusarium* species based on morphology alone is challenging due to plants infected by *F. asiaticum* often being co-infected with other *Fusarium* pathogens. Therefore, the accurate and rapid detection of *F. asiaticum* plays a vital role in minimizing the risk of both FHB and its associated mycotoxin contaminations.

As PCR technology becomes faster and more efficient, strain identification is shifting from morphology to molecular diagnosis. *F. asiaticum* isolates can be distinguished from *F. graminearum* isolates using *F. asiaticum*-specific primers and agarose gel electrophoresis [7]. Furthermore, *F. asiaticum* may be discovered precisely using real-time PCR and droplet digital PCR methods [10]. On the other hand, thermal amplification still requires expensive equipment and a lengthy process. Isothermal amplification technology, such as loop-mediated isothermal amplification (LAMP), has helped to tackle this problem and become a powerful tool because of its simplicity and sensitivity in detection [11]. However, the LAMP technique has intrinsic limitations such as cross-contamination, the need for high-quality DNA, and the inability to reuse the result [12]. Recombinase polymerase amplification (RPA) is another highly sensitive and selective isothermal amplification method, which works at 37–42 °C and can amplify as few as 1–10 DNA target copies in less than 20 min [13]. When combined with CRISPR/Cas technology, it allows more sensitive nucleic acid detection [14,15]. LbaCas12a (Cpf1) protein binds to double-stranded DNA with CRISPR RNA (crRNA) and shows non-target DNase activity on single-stranded DNA (ssDNA). RPA amplicons with protospacer-adjacent motif (PAM) site (5′-TTTN-3′) serve as targeted double-stranded DNA during the detection process. Then, for lateral flow detection (LFD), ssDNA with the sequence of 5′-FAM-TTATT-biotin-3′ is cleaved in a non-targeted way [16]. In contrast to the fluorescence readout, which can be used to visualize the results but requires equipment, the visual readout performs like LFD and can be performed in the field without the need for any sophisticated laboratory equipment [17].

In this study, a new method using the target gene *CYP51C* (the third subunit of the cytochrome P450 lanosterol C-14α-demethylase gene) is developed for the rapid detection of *F. asiaticum* based on rapid DNA extraction, recombinase polymerase amplification, CRISPR/Cas12a cleavage, and LFD. The method was also validated on wheat and maize samples and achieved true-positive results. Moreover, the entire diagnostic process was completed within 73 min, by directly detecting the *F. asiaticum* from the infected plant tissues without requiring complicated instruments or technical staff.

## 2. Results

### 2.1. Schematic Diagram of F. asiaticum Detection Based on RPA-Cas12a-LFD Assay

The principle of the CRISPR/Cas12a-based *F. asiaticum* detection system is shown in Figure 1. Initially, DNA was extracted from samples using a 5% chelex-100 solution, wherein two incubations are performed at 65 °C, and the DNA-containing supernatant was obtained after centrifugation. Subsequently, the target gene fragment was amplified by an RPA reaction. Under a constant temperature of 37 °C, the amplification process involved the formation of protein–DNA complexes between DNA recombinase and specific nucleic acid amplification primers. This complex searches for and amplifies target sequences in the *F. asiaticum* genome within the detection system, leading to exponential target gene amplification within 20 min. In the next step, the Cas12a-crRNA complex was added to the amplification system. The crRNA directs the binding of Cas12a to the RPA products by complementary base pairing. Upon combination and recognition with the PAM-containing sequence, the Cas12a endonuclease was activated, resulting in the trans-cutting of non-specific single-stranded DNA-fluorophore-biotin (ssDNA-FB) reporters in the system. Consequently, two bands appear on the lateral flow strip assay (Figure 1).

### 2.2. Optimal Reaction Conditions of RPA-Cas12a-LFD Assay for F. asiaticum Detection

RPA primers were manually designed from the *CYP51C* gene of *F. asiaticum* using the RPA primer design guidelines (Appendix A). Multiple sequence alignment was used to validate the specificity of the chosen region for *F. asiaticum*. Following that, crRNA candidates for *F. asiaticum* were designed utilizing common crRNA design methods (Figure 2). To determine the optimal reaction conditions for the RPA assay of *F. asiaticum*, several variables including reaction times, amplification temperatures, and concentrations of ssDNA-FB reporter and crRNA were tested. The genomic DNA of *F. asiaticum* was used as a constant template at a concentration of 100 ng/μL. Optimization of reaction time revealed that RPA products were visible on the test line for 10, 15, 20, and 30 min. However, when the reaction time was reduced to 10 min, the RPA product remained visible on the test line (Figure 3A). Thus, 10 min was determined to be the optimal reaction time. The effect of temperature on RPA reactions was investigated by conducting the assay at four different temperatures (35 °C, 37 °C, 39 °C, and 41 °C) for 20 min. Clear test lines were observed at all four temperatures, suggesting that the reaction temperature could range from 35 °C to 41 °C in the developed RPA assay. There were no obvious changing patterns observed concerning the changes in the temperature (Figure 3B). For the optimization of the concentrations of ssDNA-FB reporter, the reaction mixtures were incubated at 37 °C with five different concentrations (200, 300, 400, 500, and 600 nM) of ssDNA-FB reporter. The results show that the RPA product exhibited a weak signal intensity at 200 or 300 nM ssDNA-FB reporter, in contrast to higher concentrations. Therefore, the concentration of the ssDNA-FB reporter was determined to be at least 400 nM (Figure 3C). Finally, the concentrations of crRNA were optimized by culturing the reaction mixture at 37 °C with varying concentrations of crRNA (50, 100, 200, 300, and 400 nM). The results indicate that the LFD results show a weak signal on the test line at 50 nM crDNA in comparison to higher concentrations. Thus, the optimal concentration of crRNA was determined to be at least 100 nM (Figure 3D).

### 2.3. Specificity and Sensitivity Evaluation of the RPA-Cas12a-LFD Assay for F. asiaticum Detection

To further evaluate the specificity of the RPA-Cas12a-LFD assay, we extracted DNA from the *F. temperatum*, *F. boothii*, *F. oxysporum*, *F. verticillioides*, *F. graminearum*, *F. cortaderiae*, *F. proliferatum*, *F. fujikuroi*, *F. andiyazi*, *F. avenaceum*, *F. meridionale*, *F. solani*, *F. subglutinans*, *F. equiseti*, *Aspergillus niger*, *A. costaricensis*, and *Alternaria* sp. Our results show that positive signals could be only detected in the *F. asiaticum* samples within 73 min, despite the higher amount of template DNA in the control group compared to *F. asiaticum* (Figure 4). It is worth noting that our detection system only targets *F. asiaticum*, rendering it incapable of detecting genomic nucleic acid sequences of other *Fusarium* species. Hence, the CRISPR/Cas12a detection system for *F. asiaticum* demonstrated exceptional specificity. Furthermore, we estimated the sensitivity of the *F. asiaticum* RPA-Cas12a-LFD assay using varied concentrations of *F. asiaticum* genomic DNA (Figure 5). The test was successful in identifying a clear detection line at a DNA concentration of 20 fg. However, with decreasing DNA concentrations, the color of the test line also faded, and no test line appeared at DNA concentrations below 20 ag. In conclusion, the developed assay can effectively detect target DNA concentrations as low as 20 fg (Figure 5).

### 2.4. Early Diagnosis of F. asiaticum Infection in Wheat and Maize

The feasibility and validity of the RPA-Cas12a-LFD method for the detection of *F. asiaticum* in wheat and maize samples were evaluated using wheat spikes and ten-leaf-stage maize stalks pre-inoculated with *F. asiaticum*. Crude DNA was then extracted from each sample with 5% chelex-100 solution. Upon testing the samples, clear bands were visible in all inoculated samples, whereas non-inoculated samples displayed negative results (Figure 6).

Furthermore, we investigated the potential of the assay for the early detection of *F. asiaticum* infection in wheat and maize kernels. We recorded the band patterns of infected samples starting from 24 h and continuing until 120 h. As illustrated in Figure 7A, the mock samples did not display any detection lines. However, the infected wheat and maize kernels showed a weak test band after 72 h compared to the mock samples (Figure 7B,C).

## 3. Discussion

FHB is the most important wheat disease and proves challenging to prevent [18,19]. *F. asiaticum* is also a major causal agent of FHB [6]. However, distinguishing *F. asiaticum* from other *Fusarium* species linked to head blight based on morphological and cultural features is difficult. Therefore, effective, precise, field-deployable, and point-of-care rapid detection assays are urgently needed to bypass laboratory-based testing and alleviate the laborious genotyping and phenotyping of abundant samples in an extended process. The detection of *F. asiaticum* using a LAMP assay has been developed [11], requiring at least four primer pairs and an amplification time of 1 h. LAMP provides various advantages over conventional and real-time PCR, although it still requires temperature control equipment to execute the reaction at 65 °C [20]. RPA, an isothermal nucleic acid amplification methodology for detecting plant diseases, has been developed as an alternative DNA amplification method that addresses the limitations of designing LAMP primers [21]. RPA involves 30–35 bp primers forming a complex with a recombinase enzyme that binds to homologous DNA sequences. RPA assays are faster, more specific, easier, and more sensitive than LAMP procedures. RPA assays, for example, amplify DNA at constant and low temperatures, avoiding the need for thermal cyclers, and the entire detection procedure may be completed in 35 min, which is twice as quick as the LAMP technique [22]. Additionally, lateral flow dipsticks (LFDs) can be used to visualize RPA amplicons using an oligonucleotide probe in real time [23]. The rapid detection of *Fusarium* using an RPA-based fluorescence detection method has also been established [17], although we found that it is not convenient to carry a fluorescence detector with a high-power mobile battery and perform the experiments in the field. Hence, we turned our focus toward lateral flow assays, one of the most convenient testing techniques widely used for pathogen detection [24,25]. The emergence of the CRISPR/Cas system heralds a new era of detection in genetic testing [15]. This system comprises various Cas proteins with nuclease activity, which are used to detect nucleic acids [26]. Among them, Cas12a has the ability of dsDNA recognition and non-specific ssDNA collateral cleavage [14]. This ssDNA can be modified using small molecular groups such as FAM, biotin, BHQ1, and others [27,28]. The released FAM or biotin can then be detected using fluorimeters or lateral flow assays [26]. In this study, we employed FAM-TTATT-biotin as the ssDNA-FB reporter of combined RPA amplification and Cas12a-based detection to establish a portable lateral flow assay for *F. asiaticum*, to reduce the reliance on equipment, simplify the operations, and avoid contamination.

The proposed detection protocol for *F. asiaticum* requires minimal equipment, including pipettes, reagent tubes, a portable heating block, and lateral flow strips. This method has the potential to facilitate the on-site detection of plant pathogens outside of laboratories. First of all, the *CYP51C* gene is a reliable marker to resolve interspecific phylogenetic relationships within the *F. graminearum* species complex [29]. For the detection of *F. asiaticum*, we developed an RPA-Cas12a-LFD assay, where the design of specific primers is critical. The nucleotide sequence of the *CYP51C* gene varies widely among *Fusarium* spp. The optimized RPA reaction conditions of *F. asiaticum* were implemented, and we found that only 10 min was required for the RPA reaction, with reaction temperature varying from 35 °C to 41 °C. Thus, we can complete the RPA reaction at body temperature, without the need for heating equipment. Regarding ssDNA-FB reporter optimization, our results indicate that the concentration of ssDNA-FB reporter needs to be at least 400 nM. For the concentration of crRNA optimization, these results indicate that the concentration of crRNA needs at least 100 nM. Optimizing the RPA conditions of *F. asiaticum* can make our detection cheaper and faster. Moreover, one crucial step in quick detection is the genomic DNA extraction of diseased samples. Traditional genomic DNA extraction methods or kits are time-consuming and expensive. In this study, the extraction of DNA was carried out by using a 5% chelex-100 solution followed by incubation at 65 °C for 20 min, and crude DNA was collected from the supernatant after centrifugation [30,31]. This method saves at least half the time compared to the traditional method, although the concentration and purity of DNA from rapid extraction is lower than that of kits. Nonetheless, a lower concentration and purity of DNA obtained from rapid extraction is not a limiting factor in the RPA-Cas12a-LFD system, since it proves to be highly sensitive. In our study, the developed assay successfully detected DNA concentrations as low as 20 ag.

The specificity of the RPA-Cas12a-LFD assay is guaranteed by Cas12a-crRNA targeting, where LbaCas12a-crRNA can discriminate single-nucleotide mutations such as point mutations in the seed region. Our study found that the developed RPA-Cas12a-LFD assay did not cross-react with nucleic acid samples from other *Fusarium* spp., making it superior to previously reported methods. Additionally, the assay achieved successful detection in artificially inoculated samples, indicating its potential application in fields and quarantine.

## 4. Materials and Methods

### 4.1. Source of Strain Preparation and DNA Extraction

The stored fungal strains were cultured on potato dextrose agar medium plates and incubated at 28 °C for 7 days in the dark. For crude DNA extraction, the hyphae were scraped using toothpicks into tubes containing 100 µL of 5% chelex-100 solution (Chelex^®^ 100 sodium form, Sigma-Aldrich, Darmstadt, Germany) and incubated at 65 °C for 20 min. The crude DNA was collected from the supernatant and used for the next RPA reaction and detection. For sensitivity detection, genomic DNA was extracted using a TaKaRa MiniBEST Universal Genomic DNA Extraction Kit Ver.5.0 according to the manufacturer’s instructions. The quality and quantity of the extracted DNA were assessed by using P200/P200+ microvolume spectrophotometers (Pluton Technology, San Jose, CA, USA). The extracted DNA was stored at −80 °C until further use.

### 4.2. RPA Primers and CRISPR RNA Design for F. asiaticum Detection

In previous studies, the *CYP51C* gene, the third subunit of the cytochrome P450 lanosterol C-14α-demethylase gene, was selected for LAMP-based detection [11]. In this study, the same gene was selected as a target to develop the detection assay. The principle of designing RPA primers for *F. asiaticum* referred to the TwistAmp Assay Design Manual (https://www.twistdx.co.uk/wpcontent/uploads/2021/04/twistamp-assay-design-manual-v2-5.pdf, accessed on 27 July 2022) using the Snapgene software 7.0 (https://www.snapgene.com, accessed on 27 July 2022). Subsequently, crRNA candidates for *F. asiaticum* were designed using common principles of crRNA design. Synthesis of oligonucleotides and structural prediction and calculation of ΔG were performed using the mFold webserver (http://www.unafold.org/mfold/software/download-mfold.php, accessed on 27 July 2022) and OligoAnalyser (https://eu.idtdna.com/calc/analyzer, accessed on 27 July 2022). The partial *CYP51C* sequences of different strains were downloaded from NCBI, and conserved regions were chosen through multiple sequence alignment analysis. The accession numbers of different species are as follows: *F. asiaticum* (CP088259.1), *F. ussurianum* (KC020148.1), *F. vorosii* (GU785066.1), *F. pseudograminearum* (CP102999.1), *F. culmorum* (GU785033.1), *F. mesoamericanum* (GU785042.1), *F. cerealis* (GU785029.1), *F. graminearum* (LT222055.1), *F. avenaceum* (KY471127.1), *F. venenatum* (GU785060.1), *F. sporotrichioides* (GU785052.1), *F. poae* (GU785045.1), and *F. oxysporum* (XM_031192208.1). The specificity of the selected position for *F. asiaticum* was validated through multiple sequence alignment.

### 4.3. Optimizing Reaction Conditions of RPA-Cas12a-LFD Assay for F. asiaticum Detection

RPA primers were manually designed from the *CYP51C* gene of *F. asiaticum* using the RPA primer design guidelines (Appendix A). Multiple sequence alignment was used to validate the specificity of the chosen region for *F. asiaticum*. Following that, crRNA candidates for *F. asiaticum* were designed utilizing common crRNA design methods. RPA reactions were conducted using the TwistAmp Basic kit (TwistDx) according to the manufacturer’s instructions. The master mix was prepared by combining 29.5 μL TwistAmp RPA buffer, 2.4 μL forward primer (10 μM), and 2.4 μL reverse primer (10 μM), followed by 11.2 μL nuclease-free water, in tubes with pre-existed RPA enzyme. To five aliquots of the master mix, 0.4 μL of DNA template and 14 mM of Mg(OAc)_2_ were added to achieve a final reaction volume of 10 μL. The vial was immediately mixed by vigorous inversion and incubated at 37 °C for 20 min in a thermal cycler, after which the amplicon was immediately detected with the Cas12a cleavage assay. Reaction conditions, including reaction time, amplification temperature, ssDNA-FB reporter concentration, and crRNA concentration, were optimized to improve assay speed, accuracy, and cost-effectiveness. Firstly, the reaction mixture was cultured at 37 °C for six different times (10 min, 15 min, 20 min, 25 min, 30 min, and 35 min). Secondly, the reaction mixture was incubated at four different temperatures, i.e., 35 °C, 37 °C, 39 °C, and 41 °C, for 20 min. Thirdly, the reaction mixture was cultured at 37 °C for six different concentrations (200 nM, 300 nM, 400 nM, 500 nM, and 600 nM) of the ssDNA-FB reporter. Lastly, the reaction mixture was cultured at 37 °C for five different concentrations (50 nM, 100 nM, 200 nM, 300 nM, and 400 nM) of crRNA. In the final step, LFD was utilized to detect RPA amplification products by using Tiosbio^®^ Cas 12/13 Dedicated Nucleic Acid Test Strips (JY0301, Beijing Baoying Tonghui Biotechnology Co., Ltd., Beijing, China), as previously described.

### 4.4. CRISPR/Cas12a-Based Lateral Flow Strip Assay

A CRISPR/Cas12a-based reaction system composed of 50 nM LbaCas12a (M0653T, NEB), 62.5 nM crRNA, 500 nM ssDNA reporter, 2 μL of RPA product, and 2 μL of 10 × NEBuffer 2.1 (B7203S, NEB) was adjusted to 20 μL using nuclease-free water in each Cas12a reaction. Before cleavage, LbaCas12a and crRNA were incubated with 10 × NEBuffer 2.1 at 37 °C for 10 min to form the Cas12a/crRNA complex. Followed by 17 μL Cas12a/crRNA complex, 2 μL RPA products and 1 μL of ssDNA reporter were added and incubated at 37 °C for 30 min.

To perform the CRISPR/Cas12a-based lateral flow strip assay, 30 μL ddH_2_O was added to the CRISPR reactions, and the sticks were dipped into the solution for 3 min. The amplified product was recognized by biotin-ligand and anti-FAM monoclonal antibodies on the test line, where gold nanoparticles had already been fixed. The sticks were labeled with two bands, the test line and the control line, the appearance of two bands indicated the positive result.

### 4.5. Specificity and Sensitivity of RPA-Cas12a-LFD Assay for F. asiaticum Detection

Different DNA samples were taken from various strains of the *Fusarium* and non-*Fusarium* species described in Appendix A to test the system’s specificity. The RPA-Cas12a-LFD assays were evaluated to detect the *CYP51C* gene using the crude DNA from 17 isolates of *Fusarium* and non-*Fusarium* species. To evaluate the sensitivity of the developed assay, different concentrations of genomic DNA of *F. asiaticum*, i.e., 2 ng/μL, 200 pg/μL, 20 pg/μL, 200 fg/μL, 20 fg/μL, 2 fg/μL, 200 ag/μL, 20 ag/μL, and 2 ag/μL, were tested. The genomic DNA was serially diluted by adding ddH_2_O and the LFD results were observed by using the optimized assay.

### 4.6. Detection of F. asiaticum-Infected Wheat and Maize Samples by RPA-Cas12a-LFD Assay

To ensure the application of the developed assay in field conditions, *F. asiaticum* was artificially inoculated into wheat spikes and kernels as well as maize stalks and kernels. Wheat (*Triticum aestivum*) cultivar Zhongyuan 98–68 was used for spike and kernel infection, respectively. Maize (*Zea mays*) cultivar B73 was used for stalk and kernel infection assays.

For wounded wheat spikes, a 10 µL droplet of *F. asiaticum* conidial suspension having a concentration of 1 × 10^6^ conidia/mL was inoculated into three synchronous wheat flowering spikes at mid-anthesis. After inoculation, the wheat spikes were kept humid with water-soaked 9 cm qualitative filter papers (Newstar, Hangzhou, China) in plastic boxes. Pictures of the diseased samples were taken 5 days post inoculation (dpi) at 25 °C before they were harvested for further study [32]. Maize stalks were first injected with toothpicks and then inoculated with the conidial suspension at the ten-leaf stage and wrapped with soaked gauze for one week. Three replications of each experiment were conducted.

For wheat or maize kernels, 10 μL of *F. asiaticum* above conidial suspension was inoculated into the sterilized kernels and cultured at 25 °C with water-soaked 9 cm qualitative filter papers. Wheat and maize kernels were sterilized with 75% ethyl alcohol for 1 min, followed by 6% sodium hypochlorite for 3 min. Then, kernels were washed three times with distilled water before being pierced with a sterile needle to facilitate pathogen entry. The infected wheat and maize samples were used for crude DNA extraction and ground maize and wheat sample powder was scraped using toothpicks into the tube containing 100 µL of 5% chelex-100 solution and incubated at 65 °C for 20 min. The supernatant containing crude DNA was used for RPA reaction and detection. The amplified products were visually detected by the LFD developed during this study.

## 5. Conclusions

In conclusion, the RPA-Cas12a-LFD-based system proves promising for detecting *F. asiaticum* with high sensitivity and specificity, without requiring scientific instruments or skilled operators. The lyophilized RPA-Cas12a-LFD assay is now ready for industrial manufacturing and wide-ranging applications.

## Figures and Tables

**Figure 1 ijms-24-14134-f001:**
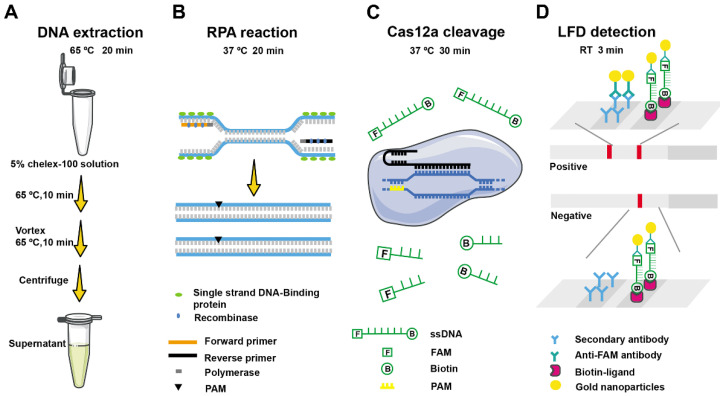
Workflow of the developed RPA-Cas12a-LFD detection system for *F. asiaticum*. (**A**) Crude DNA extraction using 5% chelex-100 solution can be completed in 20 min. (**B**) Workflow of recombinase polymerase amplification. (**C**) Cas12a cleavage. (**D**) Lateral flow detection at room temperature in 3 min. Extracted DNA or lysed samples can be used as an input for RPA-Cas12a-based detection.

**Figure 2 ijms-24-14134-f002:**
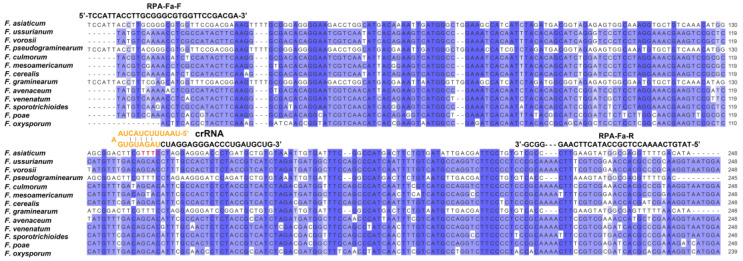
Alignment of *FaCYP51C* gene with its homologs, the design position of RPA primers, the PAM site labeled with red color (TTTC), and crRNA sequences. The crRNA contains a 21-nt direct repeat scaffold labeled with orange color and a 21-nt spacer (guide sequence).

**Figure 3 ijms-24-14134-f003:**
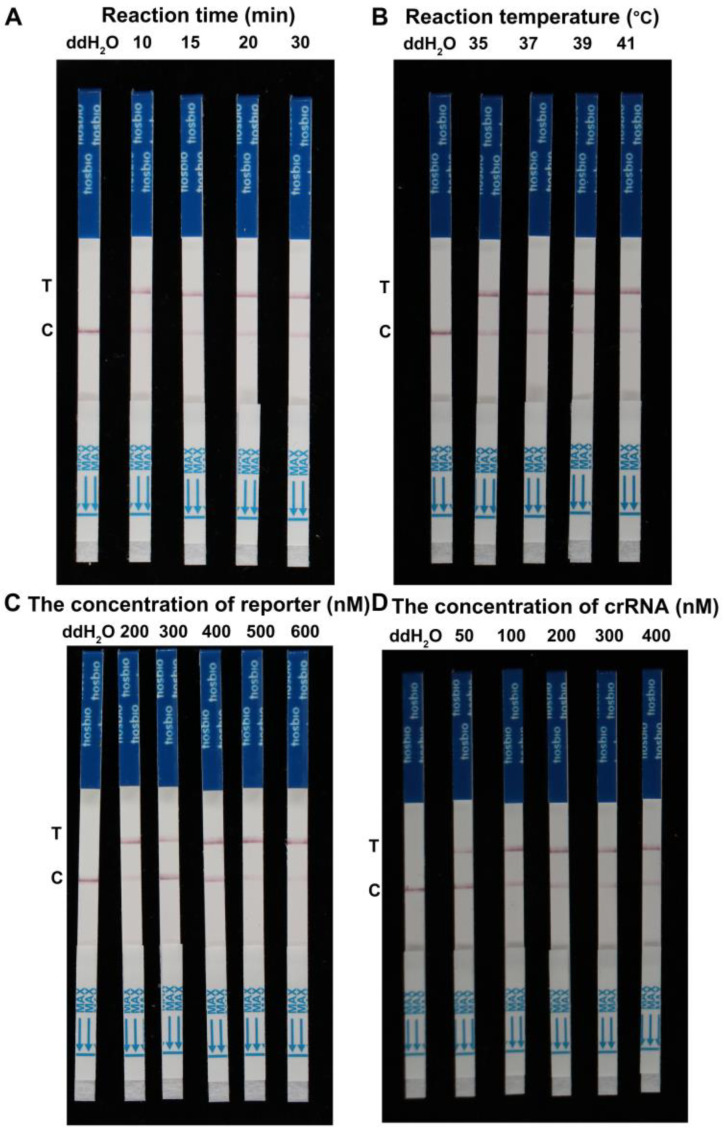
Optimization of reaction conditions for the RPA-Cas12a-LFD assay. (**A**) Optimization of the reaction volume. The RPA-Cas12a-LFD worked effectively in a broad range of reaction volumes, and 10 μL was the minimum volume required for the RPA reaction. (**B**) Determination of the reaction temperature. Clear test lines were visible in the range of 35 °C and 41 °C, and 35 °C was the minimum temperature required for the RPA reaction. (**C**) Optimization of the concentration of ssDNA-FB reporter. (**D**) Optimization of the concentration of crRNA.

**Figure 4 ijms-24-14134-f004:**
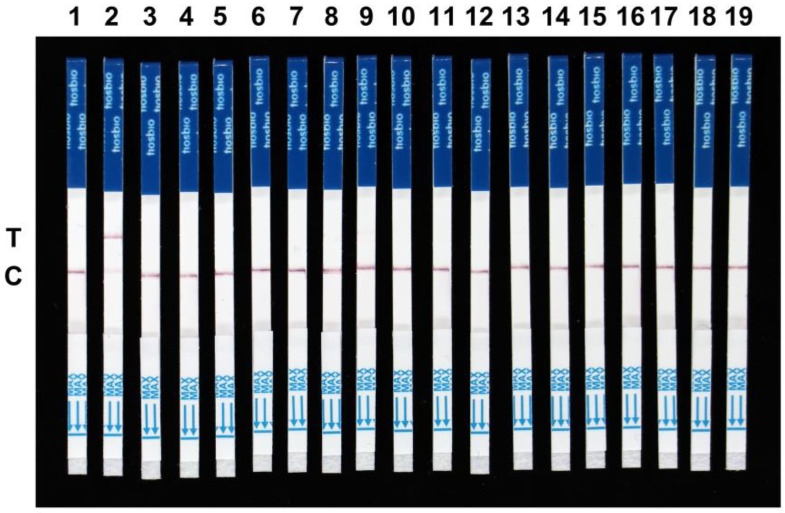
Specificity evaluation of the RPA-Cas12a-LFD detection system. Lane 1, negative control, Lane 2, *F. asiaticum*, Lane 3, *F. temperatum*, Lane 4, *F. boothii*, Lane 5, *F. oxysporum*, Lane 6, *F. verticillioides*, Lane 7, *F. graminearum*, Lane 8, *F. cortaderiae*, Lane 9, *F. proliferatum*, Lane 10, *F. fujikuroi*, Lane 11, *F. andiyazi*, Lane 12, *F. avenaceum*, Lane 13, *F. meridionale*, Lane 14, *F. solani*, Lane 15, *F. subglutinans*, Lane 16, *F. equiseti*, Lane 17 *A. niger*, Lane 18, *A. costaricensis*, and Lane 19, *Alternaria* sp. RPA was carried out in a 10 μL reaction mixture per sample containing 0.2 μL genomic DNA (100 ng/μL) as a template.

**Figure 5 ijms-24-14134-f005:**
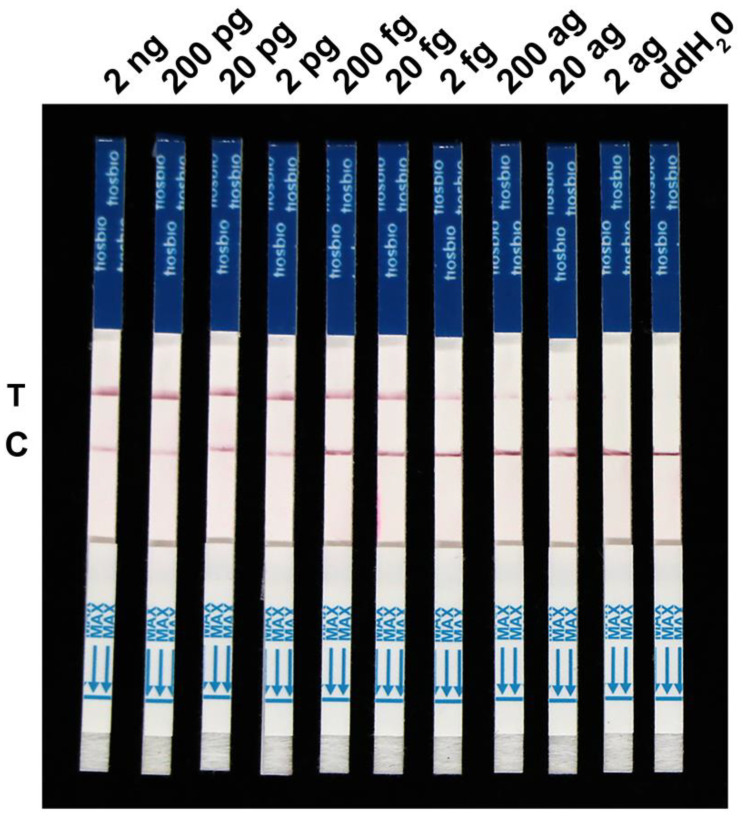
Sensitivity evaluation of the RPA-Cas12a-LFD detection system. The sensitivity of detection of the developed RPA-Cas12a-LFD assays was determined using serially diluted genomic DNA of *F. asiaticum* from 2 ng to 2 ag.

**Figure 6 ijms-24-14134-f006:**
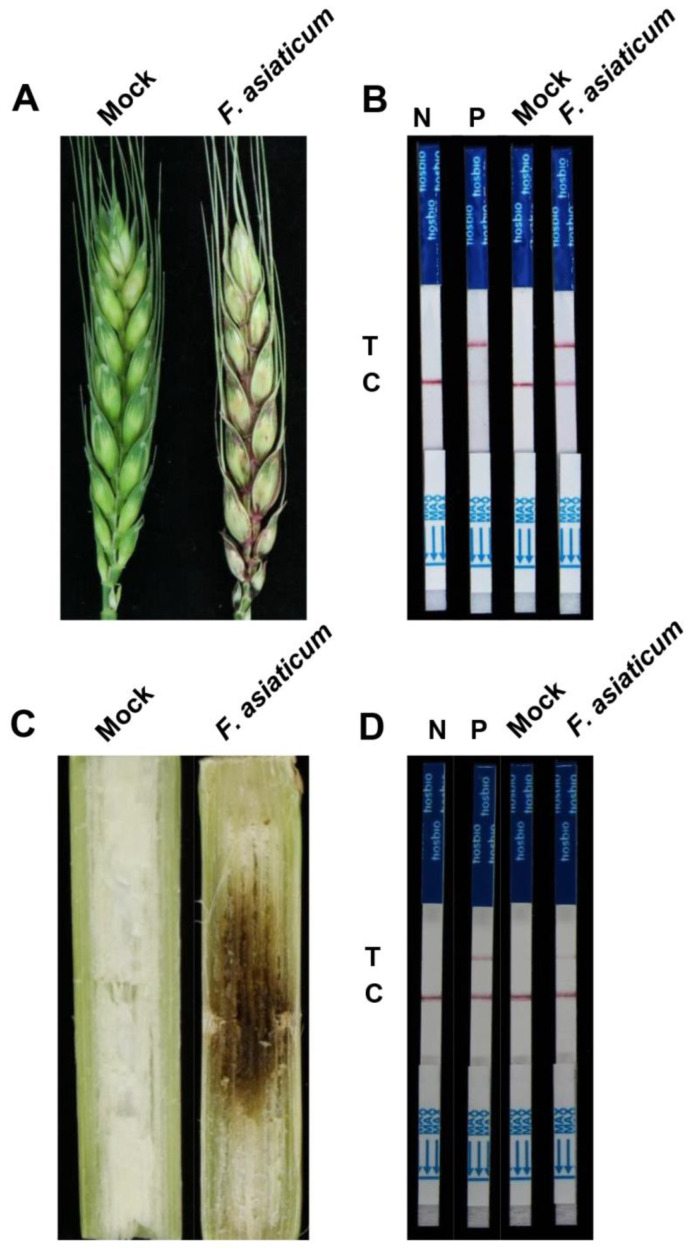
Detection of *F. asiaticum* in diseased wheat spikes and maize stalks using the developed RPA-Cas12a-LFD assay. (**A**) Symptoms of wheat spikes inoculated with *F. asiaticum* at 5 days post inoculation. (**B**) Detection and identification of *F. asiaticum* in diseased wheat spikes using the developed RPA-Cas12a-LFD assay. (**C**) Symptoms of maize stalks inoculated with *F. asiaticum* at 7 days post inoculation. (**D**) Detection and identification of *F. asiaticum* in diseased maize stalks using the developed RPA-Cas12a-LFD assay. The positive control (P) consisted of 100 ng/μL genomic DNA from *F. asiaticum*, while the negative control (N) was represented by ddH_2_O. Additionally, genomic DNA extracted from wheat spikes (**B**) or maize stalks (**D**) inoculated with ddH_2_O (Mock) and *F. asiaticum*, respectively, served as testing samples.

**Figure 7 ijms-24-14134-f007:**
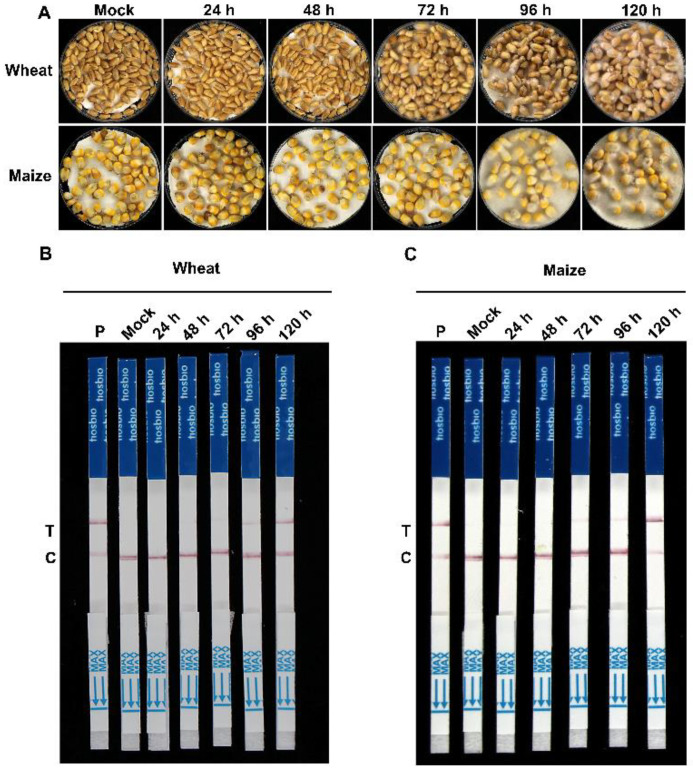
Detection of *F. asiaticum* in diseased wheat and maize kernels using the developed RPA-Cas12a-LFD assay. (**A**) Symptoms of wheat and maize kernels inoculated with *F. asiaticum* at 24, 48, 72, 96, and 120 h post inoculation. Detection and identification of *F. asiaticum* in diseased wheat (**B**) and maize (**C**) kernels using the developed RPA-Cas12a-LFD assay. The positive control (P) consisted of 100 ng/μL genomic DNA from *F. asiaticum*. Genomic DNA extracted from wheat (**B**) or maize (**C**) kernels inoculated with ddH_2_O (Mock) and *F. asiaticum*, respectively, served as testing samples.

## Data Availability

The original contributions presented in the study are included in the article. Further inquiries can be directed to the corresponding author.

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
