# Peer review of "Rapid and Sensitive Detection of Toxigenic Fusarium asiaticum Integrating Recombinase Polymerase Amplification, CRISPR/Cas12a, and Lateral Flow Techniques"

_ijms, 2023, doi:10.3390/ijms241814134_

Round 1
Reviewer 1 Report
This is a carefully done study and the findings are of considerable interest. A few minor
revisions are listed below.
Tables S1 and S2 are not attached to this document.
Figure 1: Although it can be roughly guessed, it is not clearly indicated which part of the figure corresponds to (A), (B), (C), or (D).
Page 3 Line 2, 4, 5, 6, 8: F. asiaticum should be Italic.
Page 3 Line 5-6: (Figure 2A) should read (Figure 2).
Page 4: Figure 2 should be larger in size.
Figure 2: The RPA-Fa-F primer sequence is aligned with the sequence of F. asiaticum, but there is a slight 5 base deviation (TCCAT) at the 5' end.
Page 4: Figure 2 requires annotation. Each sequence in Figure 2 should reveal the accession number.
Page 4: The yellow-green border enclosure in Figure 2 should be described.
Wouldn't it be advisable to omit the 3-base sequence 'CGG' from the second to fourth positions at the 3' end of the RPA-Fa-R primer (3'-GCGGGAACTTCATACCGCTCCAAAACTGTAT-5')? Such a 3-base sequence is not present in the corresponding region of F. asiaticum (Figure 2). I would appreciate a clear explanation regarding this matter.
Page 5 Line 4-7: “Fusarium temperatum”, “F. boothii”, “F. oxysporum”, “F. verticillioides”, “F. graminearum”, “F. cortaderiae”, “F. proliferatum”, “F. fujikuroi”, “F. andiyazi”, “F. avenaceum”, “F. meridionale”, “F. solani”, “F. subglutinans”, “F. equiseti”, “Aspergillus niger”, “A. costaricensis” and “Alternaria” should be Italic.
Page 5 Line 7-10, 12-14: F. asiaticum should be Italic.
Page 5 Line 15: (Figure 5A and B) should read (Figure 5)
Page 5 Line 9: (Figure 4A and B) should read (Figure 4)
Page 6 Line 4, 8: F. asiaticum should be Italic.
Page 9 Line 46: (Xu et al., 2017) should read [12].
Figure 3: “35 â—¦C and 41 â—¦C, and 35 â—¦C” should read “35 °C and 41°C, and 35 °C”.
Figure ï¼”: “F. Solani” should read “F. solani”.
The source of purchase for the lateral flow strip is not specified.
Author Response
We would like to express our sincere gratitude to you and all the reviewers for their valuable comments and professional advice regarding our paper. Your opinions have been immensely helpful in improving the academic quality of our work. ​After careful consideration of your suggestions, we have made all corrections to the revised manuscript. These changes have been highlighted with red color. A detailed description of all the modifications has been attached. We sincerely hope that our responses and revisions meet your expectations and that the paper is now suitable for publication.

Reviewer 2 Report
I congratulate the authors for their work and manuscript. It is very pleasing to see the development of applications to molecular knowledge. The authors employed a combination of molecular methods to develop a portable assay for Fusarium Head Blight. The lateral flow assay developed herein has the advantage of being highly sensitive and specific to F. asiaticum while at the same time being simple enough to perform without training or sophisticated equipment. It is a great contribution to prevent spreading of this important disease. The manuscript is well written and I have no issues before recommending acceptance for publication in IJMS.
Unfortunately the line numbers were not included so I'll refer to page and section numbers only on my comments.
Page 2, 2nd paragraph: "In comparison, the visual readout outperforms like LFD can be per-formed in the field, and does not require any laboratory equipment." Review this sentence for clarity.
Make all mentions of F. asiaticum and other scientific names in italic.
Author Response

(The authors gave the same response as above.)

Reviewer 3 Report
Dear Authors
The present manuscript entitled “Rapid and sensitive detection of toxigenic Fusarium asiaticum integrating recombinase polymerase amplification, CRISPR/Cas12a and lateral flow techniques” discuss the new and efficient diagnosis method of F. asiaticum infection in maize and wheat kernels even in 3 days of inoculation and suggest that the developed assay will provide an efficient and robust detection platform to accelerate plant pathogen detection. The more general question regarding this work is; present method claims to be very specific for F. asiaticum and even did not detect the other Fusarium species. More prevalent infection in Maize crop is F. verticilliodes in Mezican fields which is major corn producer in the world. The present method is then useful for specific regions of Asian continent where F. asiaticum is prevalent; please explain.
In the methods section “10 µL droplet of F. asiaticum conidial suspension was inoculated into wheat spikes at mid-flowering times” is claimed. Please include the disease development stages, appearance of first symptoms and where the plants were maintained etc? The discussion need to be more elaborated with more and recent references.
Thank you
Regards
Author Response

(The authors gave the same response as above.)
